# Targeting MHC Regulation Using Polycyclic Polyprenylated Acylphloroglucinols Isolated from *Garcinia bancana*

**DOI:** 10.3390/biom10091266

**Published:** 2020-09-02

**Authors:** Chloé Coste, Nathalie Gérard, Chau Phi Dinh, Antoine Bruguière, Caroline Rouger, Sow Tein Leong, Khalijah Awang, Pascal Richomme, Séverine Derbré, Béatrice Charreau

**Affiliations:** 1Université de Nantes, CHU Nantes, Inserm, Centre de Recherche en Transplantation et Immunologie, UMR 1064, ITUN, F-44000 Nantes, France; chloe.coste@agrocampus-ouest.fr (C.C.); Nathalie.Gerard@univ-nantes.fr (N.G.); 2SONAS, EA921, University of Angers, SFR QUASAV, Faculty of Health Sciences, Department of Pharmacy, CEDEX 01, 49045 Angers, France; chauphi.dinh@univ-angers.fr (C.P.D.); antoine.bruguiere@outlook.com (A.B.); caroline.rouger@u-bordeaux.fr (C.R.); pascal.richomme@univ-angers.fr (P.R.); 3Department of Chemistry, Faculty of Science, University of Malaya, Kuala Lumpur 50603, Malaysia; sowtein@hotmail.com (S.T.L.); khalijah@um.edu.my (K.A.)

**Keywords:** endothelium, Clusiaceae, *Garcinia bancana*, guttiferone F, guttiferone J, major histocompatibility complex, HLA-E, polycyclic polyprenylated acylphloroglucinols, xanthochymol, histone acetyltransferase

## Abstract

Modulation of major histocompatibility complex (MHC) expression using drugs has been proposed to control immunity. Phytochemical investigations on *Garcinia* species have allowed the isolation of bioactive compounds such as polycyclic polyprenylated acylphloroglucinols (PPAPs). PPAPs such as guttiferone J (**1**), display anti-inflammatory and immunoregulatory activities while garcinol (**4**) is a histone acetyltransferases (HAT) p300 inhibitor. This study reports on the isolation, identification and biological characterization of two other PPAPs, i.e., xanthochymol (**2**) and guttiferone F (**3**) from *Garcinia bancana*, sharing structural analogy with guttiferone J (**1**) and garcinol (**4**). We show that PPAPs **1**–**4** efficiently downregulated the expression of several MHC molecules (HLA-class I, -class II, MICA/B and HLA-E) at the surface of human primary endothelial cells upon inflammation. Mechanistically, PPAPs **1**–**4** reduce MHC proteins by decreasing the expression and phosphorylation of the transcription factor STAT1 involved in MHC upregulation mediated by IFN-γ. Loss of STAT1 activity results from inhibition of HAT CBP/p300 activity reflected by a hypoacetylation state. The binding interactions to p300 were confirmed through molecular docking. Loss of STAT1 impairs the expression of CIITA and GATA2 but also TAP1 and Tapasin required for peptide loading and transport of MHC. Overall, we identified new PPAPs issued from *Garcinia bancana* with potential immunoregulatory properties.

## 1. Introduction

Major histocompatibility complex (MHC) molecules, also known as histocompatibility antigens or human leucocytes antigens (HLA) in humans, are highly polymorphic glycoproteins encoded by MHC class I and MHC class II genes. Classical MHC molecules are triggers of innate and adaptive immune responses against pathogens and tumors [1]. They are involved in the presentation of peptide antigens to T cells. In humans, there are three class I genes, called HLA-A, -B and -C, and three of MHC class II genes, called HLA-DR, -DP and -DQ. MHC class II molecules provide antigen presentation to induce antigen-specific CD4 T cells while MHC class I molecules trigger the activation of CD8 T cells to generate cytotoxic T lymphocytes and natural killer (NK) cells to eradicate infected or transformed cells [2]. As a part of the immune response, to enhance lymphocyte activation, expression of MHC molecules is upregulated by interferon-γ (IFNγ), which transduces signal via the Janus tyrosine kinase (Jak)1 and the latent cytosolic factor, signal transducer and activator of transcription (STAT)1. MHC class II transactivator (CIITA) is a global regulator involved in basal and IFNγ mediated MHC transcription in two distinct ways: as a transcriptional activator that nucleate an enhanceosome and as a transcription factor with acetyltransferase and kinase activities [3,4].

In addition to the highly polymorphic ‘classical’ MHC class I and class II genes, there are other genes encoding MHC class I-type, called MHC class Ib, molecules that show little polymorphism and in some cases a restricted pattern of cellular expression [5]. They include the members of the MIC gene family encoding the MHC class I chain-related A (MICA) and MICB proteins, which are under a different regulatory control from the classical MHC class I genes and are induced in response to cellular stress [6]. MICA and MICB play a part in innate immunity as ligands for the receptor NKG2D, namely NKG2DLs, expressed on NK, gamma delta T (γδ T) and CD8 T cells and enabling these cells to kill NKG2DL-expressing targets. NKG2DLs also include the UL16 binding proteins (ULBP) [7]. HLA-E is another MHC class Ib molecule being involved in both innate and adaptive immunity with a specialized role in cell recognition by NK and of CD8 T cells [8,9]. HLA-E/peptide complexes display a dual activity. They are ligands for a multimeric receptor composed of a member of the NKG2 family (NKG2A or NKG2C) complexed with CD94 on NK and on CD8 T cells as well as for T-cell receptor of a subset of CD8 T cells [10]. CD94/NKG2A engagement inhibits the cytotoxic activity of the NK and CD8 T cells. Virally infected and cancer cells manage to escape the immune responses by the negative control of classical MHC but also by the regulation and shedding of class Ib MICA, MICB and HLA-E [11]. Modulation of MHC gene expression using drugs or immunotherapies has been proposed to reinforce the immunity against viral infections and cancers or in contrast to induce immune tolerance to treat autoimmune diseases and allergies or to avoid rejection of allotransplants [12]. Recent approaches targeting specifically MIC proteins, NKG2DLs and HLA-E or their receptors to avoid the immunosuppressive action of these MHC molecules in cancer are currently under investigations [13,14,15]. During inflammation and active immune responses, proinflammatory cytokines such as tumor necrosis factor (TNF) and IFNγ are produced by immune cells and regulate the expression of genes and proteins necessary to promote adapted immune cell responses. Regulation of MHC molecules including classical class I and class II MHC (namely HLA in human) as well as non-classical MHC such as HLA-E and MICA/B on professional and no professional antigen-presenting cells such as endothelial cells (ECs) is a key process initiating both innate and adaptive immune responses.

In an attempt to identify novel immune-regulatory natural products (NPs), we recently reported on the ability of guttiferone J (**1**), a polyprenylated polycyclic acylphloroglucinol (PPAP) isolated from a *Garcinia virgata* herbal extract [16], to reduce inflammation as well as immunogenicity of the endothelium by inhibiting cytokine signaling pathways [17]. For a better understanding of underlying molecular mechanisms and targets, the present study reports on the isolation, identification and biological characterization of selected PPAPs, i.e., xanthochymol (**2**) and guttiferone F (**3**) from *Garcinia bancana*, which, by structural analogy, may have similar effects to guttiferone J (**1**) or garcinol (**4**). The functional activity of PPAPs **1**–**4** on the expression and transcriptional activation of class I and class II MHC and a set of non-classical MHC molecules such as HLA-E and MICA/B was assessed in vitro using a cellular model consisting of human primary EC cultures treated with proinflammatory cytokines to recapitulate the features of microvascular inflammation.

## 2. Materials and Methods

### 2.1. Reagents for Biological Assays

Garcinol (**4**) was purchased from Enzo Life Sciences (Villeurbanne, France), A485 was purchased from Bio-Techne (Rennes, France) and zoledronic acid (ZA), simvastatin (Sim), Vorinostat (SAHA), Trichostatin A (TSA) and C646 (4-[4-[[5-(4,5-Dimethyl-2-nitrophenyl)-2-furanyl]methylene]-4,5-dihydro-3-methyl-5-oxo-1H-pyrazol-1-yl]benzoic acid) were purchased from Sigma-Aldrich (Saint-Quentin Fallavier, France). Guttiferone J (**1**) has been kept in the in-house chemical library of our laboratory, which includes 139 polyphenols isolated from clusiaceous and calophyllaceous species. The extraction and purification of **1** have been previously described [16].

### 2.2. Plant Material

*Garcinia bancana* bark was collected in October 2000 around Mersing, Johor. The plant was identified by the botanist Mr. Teo Leong Eng and the voucher specimen (KL4967) was deposited at the Herbarium of the Department of Chemistry, University of Malaya, Kuala Lumpur, Malaysia.

### 2.3. Extraction and Purification

Bark powder (10 g) were successively extracted by sonication (3 h) with DCM and methanol (MeOH) to afford DCM (740 mg) and MeOH extracts (1.35 g), respectively. The DCM extract (650 mg) was then fractionated using normal phase flash chromatography on silica gel (Chromabond^®^ flash RS 40 SiOH) from 100% cyclohexane to 70% cyclohexane/30% ethyl acetate (flow: 20 mL/min) leading to 10 sub-fractions: F1 (15.2 mg), F2 (12.2 mg), F3 (13.2 mg), F4 (18.8 mg), F5 (30.1 mg), F6 (6.5 mg), F7 (53.7 mg), F8 (27.6 mg), F9 (22.9 mg) and F10 (25.9 mg). F7 (30 mg) was separated using semipreparative HPLC (Agilent HP 1100 Series, Agilent Technologies, Les Ulis, France) on a reverse phase column (Phenomenex Luna C18, 100 Å, 250 mm × 10 mm, 5 µm), using a 50 mg/mL concentration for the injection (100 µL), with a 97% methanol + 0.1% formic acid/3% water + 0.1% formic acid system (flow: 2.8 mL/min). Fractions were collected using the Agilent Technologies 1260 Infinity G1364C fraction collector and the ChemStation for LC 3D software for automatic UV peak detection (diode array detector G13115A). This led to 8.7 mg of xanthochymol (**2**) [18] and guttiferone F (**3**) [19] as a mixture, namely **GX**. Another semi-preparative HPLC on a PFP column (Hypersil Gold PFP, 150 mm × 10 mm), using a 50 mg/mL concentration for the injection (100 µL), with a 75% methanol/25% water + 0.1% formic acid system (flow: 4.7 mL/min) yielded 2.9 mg of pure xanthochymol (**2**) and 3.6 mg of pure guttiferone F (**3**) from the remaining F7.

### 2.4. GX Analysis and ***2***–***3*** Purity

GX and PPAPs **2** and **3** were analyzed using HPLC (Agilent HP 1100 Series) on a PFP column (Hypersil Gold PFP, 150 mm × 4.6 mm), using a 1 mg/mL concentration for the injection (20 µL), with a MeOH-H_2_O + 0.1% formic acid gradient (55%→90% MeOH at 0–60 min, 90% MeOH 60–80 min, flow: 1 mL/min; Appendix A).

### 2.5. NMR Experiments

^1^H and ^13^C NMR spectra were recorded in methanol-d4 + 0.1% trifluoroacetic acid-d on a JEOL 400 MHz YH spectrometer (Jeol Europe, Croissy-sur-Seine, France). Chemical shifts (δ_H_ and δ_C_) are expressed in ppm and *J* values in Hz (Appendix A).

### 2.6. Cell Culture and Treatments

Human primary vascular ECs were isolated as we previously reported [9,20] and used between passages 2 and 5. ECs were cultured in an endothelial cell basal medium (ECBM) supplemented with 5% fetal calf serum (FCS), 0.004 mL/mL ECGS/heparin, 0.1 ng/mL hEGF, 1 ng/mL hbFGF, 1 µg/mL hydrocortisone, 50 µg/mL gentamicin and 50 ng/mL amphotericin B (C-22010, PromoCell, Heidelberg, Germany). ECs isolated from different donors (*n* = 5) were used in replicate experiments to ensure HLA allele diversity and avoid HLA-type-dependent effect. For activation, confluent EC monolayers were starved overnight and incubated with recombinant human IFNγ (100 U/mL, BioTechne, Abingdon, UK) for the indicated period of time in ECBM supplemented with 2% FCS. When applicable, cells were preincubated with natural or commercial compounds (1–20 µM) or diluent alone (DMSO 1/1000) for 1 h before further incubation with TNFα (100 U/mL) or IFNγ (100 U/mL) for 18 h or 48 h. Negative controls were assessed using diluent alone (DMSO 1/1000) but also structurally irrelevant biomolecules such as zoledronic acid and simvastatin.

### 2.7. Cell Immunostaining and Flow Cytometry

After treatment, cells were harvested using trypsin/EDTA before immunostaining. Cells were labeled using anti-pan HLA class I (anti-HLA-A, -B and -C; clone W6/32), anti-pan HLA class II (anti-HLA-DR, -DP, -DQ; clone L243), anti-MICA (clone AMO1; BamOmab, Tubingen, Germany), anti-HLA-E-APC (clone 3D12; Miltenyi Biotec, Paris, France) mouse IgG as primary antibodies and anti-mouse IgG + IgM (H + L)-FITC (Jackson Immunoresearch Laboratories, West Grove, PA, USA) as secondary antibodies. An isotype-matched IgG was used as the negative control. Fluorescence was measured by flow cytometry on 10,000 cells/sample using a BD FACSCanto^TM^ II flow cytometer (Becton Dickinson, Satn Jose, CA, USA). Acquired data were analyzed with FlowJo software (Tree Star, Inc., Ashland, OR, USA) and depicted in histograms plotting geometric mean of fluorescence intensity (GFI) on a four-decade logarithmic scale (*x*-axis) versus cell number (*y*-axis).

### 2.8. Cellular Viability Assay

Cell toxicity was quantified by 3-(4,5-dimethylthiazol-2-yl)-2,5-diphenyltetrazolium (MTT) colorimetric assays. ECs were plated onto 96-well plates (Nunc) precoated with 1% gelatin at 1 × 10^4^ cells/well. Confluent EC monolayers were incubated with the tested compounds in the presence of IFNγ (100 IU/mL) for 18 h. After treatment, cell viability was assessed by incubation with 1 mg/mL MTT (Sigma) for 4 h at 37 °C and recording the Optical Density at 550 nm. Experiments were performed in triplicates, and results are expressed as a percentage ± SEM values. The relationship between OD and cell number was determined to be linear by the regression curve and the equation of the curve allowed us to determine the cell number for each treatment. The relative cell viability (%) was expressed as a percentage relative to the cells treated with IFNγ and diluent (DMSO 1/1000) instead of compounds.

### 2.9. Quantitative Real-Time RT-PCR

RNA were isolated using TRIzol reagent (Invitrogen, Carlsbad, CA, USA), analyzed by the Caliper LabChip GX Analyzer (Perkin Elmer Inc., Wellesley, MA, USA) for quantity and quality, and treated with DNase (Ambion, Austin, TX, USA) before reverse transcription (RT). Quantitative PCRs (qPCRs) were performed using the ABI PRISM 7700 sequence detection application program (PE Applied Biosystems, Foster City, CA, USA). For quantification, means of C_t_ triplicates were normalized by the concomitant quantification of ribosomal protein lateral stalk subunit P0 (RPLP0, gene ID: 6175). Relative expression was calculated according to the 2^−ΔΔCt^ method, as previously described [21], and using cells treated with diluent only as “calibrators” for the relative quantification. Transcript levels were quantified by real time qRT-PCR with the following predesigned TaqMan^®^ Gene Expression Assays (FAM™ dye-labeled MGB probe), containing primers and probes, according to the manufacturer’s recommendations (Applied Biosystems, Foster City, CA, USA): Interferon Regulatory Factor 9 (IRF9 (Hs00196051_m1)), HLA-A (Hs0740413_g1), HLA-E (Hs03045171_m1), MICA (Hs00792195_m1), CIITA pIV (Hs00172106_m1), GATA2 (Hs00231119_m1), STAT1 (Hs01013996_m1), HDAC3 (Hs00187320_m1), CBP p300 (Hs00914223_m1), CREB CBP (Hs00932878_m1), SOCS1 (Hs00705164-s1), RPLPO (Hs99999902_m1), MICB (Hs00792952_m1), β2-microglogulin (Hs00984230_m1), Transporter associated with Antigen Processing 1 (TAP1(Hs00388675_m1)), TAP Binding protein (TAPBP (Hs00542606_m)), ULBP2 (Hs00607609_mHl), ULBP3 (Hs00225909_m1) and protein tyrosine phosphatase non-receptor type 11 (PTPN11 (Hs00275784_m1)).

### 2.10. Western Blot Analysis

Cells were lysed in RIPA buffer containing protease and phosphatase inhibitors (Sigma-Aldrich). Protein concentration was determined using the bicinchoninic acid (BCA) protein assay reagent (Pierce, Rockford, IL, USA). Cell lysates were resolved by SDS–PAGE (10%) and proteins were transferred to nitrocellulose membranes (ECL Hybond^TM^; Amersham, UK) using a Trans-Blot SD Semi-Dry Electrophoretic Transfer Cell (Bio-Rad, Marne-la-Coquette, France). Then, membranes were subjected to immunoblot analysis using primary antibodies and appropriate peroxidase-conjugated secondary antibodies. Primary antibodies (dilution 1/1000) were from Cell Signaling Technology (CST, Danvers, MA, USA) and directed against: GAPDH, acetylated-CBP (K1535)/p300 (K1499), STAT1 and phosphorylated-STAT1 (Y701). Antibody-bound proteins were detected using an enhanced chemiluminescence kit (West Pico ECL, Thermo Fischer) and a luminescent image analyzer LAS-4000 (Fujifilm, Tokyo, Japan). Blot and image analysis were performed with Multi Gauge^®^ (Fujifilm) and ImageJ^®^ software. Results shown are representative of at least three independent experiments.

### 2.11. Data and Statistical Analysis

Data are represented as means ± SD for replicates experiments. Statistical analyses were performed with Graphpad Prism^®^ Software (Graphpad Software, San Diego, CA, USA) by a Student’s *t*-test, a parametric or Kruskal Wallis non-parametric analysis of variance test as appropriate. A *p* value < 0.05 (*) was considered statistically significant. (**) indicates *p* < 0.01 and (***) *p* < 0.005.

### 2.12. Molecular Docking

Protein preparation: The 3D structure of histone acetyltransferase p300 in complex with a co-crystallized inhibitor was downloaded from Protein Data Bank (PDB entry 5KJ2, rcsb.org) [22]. All implicit hydrogens (of the protein and the ligand) were added using GOLD 5.6.3 (CCDC, Cambridge, UK) [23]. Water molecules that take part in at least three hydrogen bonds with the protein and/or the ligand, at least 2 of which are with the protein, were kept. Other water molecules were deleted. Hydrogen bonds were analyzed using LigandScout 4.4 (Inteligand, Vienna, Austria) [24]. Protein treatment was done with GOLD 5.6.3. The protein structure (in the absence of the ligand) was saved as a mol2 file.

Ligand preparation: Three types of ligands for redocking were prepared as follows:Co-crystallized extracted ligand from the complex structure with no changes in conformation and configuration: the ligand (after hydrogens were added) was saved as a mol2 file without the presence of the protein and any further modification.Energy-minimized co-crystallized ligand extracted from the complex: the mol2 ligand structure was energetically minimized with the built-in MMFF94 function of LigandScout 4.4 and saved as another mol2 file.Energy-minimized reconstructed ligand: the structure of the ligand was rebuilt with ChemDraw Professional 16.0 [25], its 3D structure was then visualized from the SMILES code in the ligand-based view of LigandScout 4.4, after which the energy minimization step was carried out as described above. The fully processed ligand structure was output and saved as a mol2 file.

All test-set molecules (guttiferone J (**1**), xanthochymol (**2**), guttiferone F (**3**) and garcinol (**4**)) were downloaded from SciFinder in the sd file format. Their 2D structures were then converted into 3D, and energy minimization was carried out with LigandScout 4.4 as previously described. All output ligands were saved separately as a mol2 file.

Molecular docking: A rigid molecular docking procedure was carried out using GOLD 5.6.3, with all input structures (ligands and protein) in mol2 file format. The binding site comprised of all protein residues with at least one heavy atom located within 10 Å from the centroid of the co-crystallized ligand (HET code: 6TF) whose coordinates are as follows: 34.70, 9.85 and 184.79. The CHEMPLP scoring function was used to rank the output poses (Appendix A). A maximal number of 10 poses were retained for each test set ligand. Post-docking interactions of all ligands with the protein were analyzed in LigandScout 4.4.

## 3. Results

### 3.1. Phytochemical Analysis

Natural PPAPs are usually isolated from *Garcinia* (Clusiaceae) and *Hypericum* (Hypericaceae) genera [26]. For the present study, PPAPs exhibiting a similar molecular scaffold to guttiferone J (1) were required. According to the exhaustive database created by Grossman and colleagues [27], on the 774 known naturally occurring PPAPs, 75 of them share the selected skeleton (Figure 1). Among them, garcinol (**4**) is commercially available and was thus selected for biological assays. Moreover, to date, about 75% of PPAPs exhibiting the selected scaffold were mainly biosynthesized by *Garcinia* species. The genus *Garcinia* includes about 400 trees and shrubs growing in tropical and equatorial areas including Malaysia [28,29]. Therefore, the phytochemistry of various organs (i.e., bark, leaf and sometimes fruit) of 17 *Garcinia* species (30 batches), growing in Malaysia was investigated. The apolar NPs biosynthesized by the different organs from selected samples were extracted by dichloromethane (DCM) and extracts were analyzed using HPLC coupled to UV and mass spectrometry (LC-UV-MS^2^, Appendix A). The DCM bark extract of *Garcinia bancana* was prone to contain PPAPs with the selected chemical scaffold as major compounds. After fractionation, a dereplication based on ^13^C-NMR experiments confirmed the presence of isomers of garcinol (**4**), i.e., xanthochymol (**2**) and guttiferone F (**3**) as well as derivatives [30,31]. **2** and **3** share very similar structures and were first obtained as a mixture called GX. As previously described [32,33], the isolation of such PPAPs was challenging. Using a semipreparative HPLC, a PFP column allowed the purification of **2** and **3**. Thus, the effect and mechanisms of action of PPAPs **1**–**4** were further investigated on endothelial cells that express a broad spectrum of MHC molecules including class I, class II, HLA-E, MICA and MICB [5].

### 3.2. Comparative Inhibition of MHC Molecules Mediated by Guttiferones J (***1***) and F (***3***) and Xanthochymol (***2***)

ECs were either unstimulated or stimulated with TNF or IFNγ in the presence of the mixture **GX** [**2**–**3** (4:6)], **1** or diluent (DMSO) alone as negative control. Unstimulated cells were used to define baseline levels of MHC molecules. After treatment, cells were harvested and surface expression for a panel of MHC molecules including HLA class I, HLA-class II, HLA-E and MICA was measured by flow cytometry (Figure 2A). Firstly, we found that no effect of (**2**–**3**) or (**1**) on the basal level of MHC molecules. When compared to basal levels, TNF and IFNγ induced a strong upregulation of MHC molecules compared to basal levels. HLA class II molecules were upregulated by IFNγ but not by TNF. Treatment with **GX** [**2**–**3** (4:6)] inhibited the regulation of HLA class I, HLA-Class II, HLA-E and MICA induced in response to TNF or IFNγ on ECs. An inhibitory effect was quantitatively similar and even higher for **GX** compared to **1**. Major inhibitory effects of **GX** were observed for HLA class I and HLA-E molecules. A dose-response analysis indicated that **GX** [**2**–**3** (4:6)] achieved a maximal inhibition of MHC molecules on the EC surface at 10 µM (Figure 2B). Consequently, PPAPs **2** and **3** were purified. After purification, the regulatory effect of xanthochymol (**2**) and guttiferone F (**3**) on the expression of MHC proteins was further investigated. In this functional study, **2** and **3** were compared to diluent (DMSO) alone. Cells were treated with IFNγ for 48 h and surface proteins were analyzed by flow cytometry. Figure 2C shows that **2** and **3** efficiently decreased the expression of MHC class I and class II, HLA-E and MICA at the cell surface compared to diluent and confirmed the inhibition obtained with their mixture **GX** [**2**–**3** (4:6)]. A higher rate of MHC inhibition was achieved by **3** (up to 75% of inhibition) while inhibition by **2** was moderate (25–40% of inhibition).

### 3.3. Guttiferones J (***1***) and F (***3***) and Xanthochymol (***2***) Are Novel Inhibitors of HAT CBP/p300 Activity, Which Impair IFNγ Signaling and Ultimately MHC Expression through the Inhibition of STAT1 Transcriptional Activities

To decipher the mechanisms involved in the inhibition of MHC molecules mediated by PPAPs, we employed quantitative real time RT-PCR (qRT-PCR) to quantify cellular mRNA levels and we focused on the IFNγ signaling that mediate the regulation of most MHC molecules. In these experiments, cells were treated with diluent alone, **GX** [**2**–**3** (4:6)], **1** or garcinol (**4**) in the presence of IFNγ for 18 h before cell lysis and RNA isolation. For comparison, cells were also treated with a statin (simvastatin) or with zoledronic acid (ZA), two compounds previously reported to downregulate MHC antigens [17,34]. For this study, we focused on the two MHC molecules involved in both innate and adaptive immunity, namely HLA-E and MICA. Firstly, qRT-PCR showed that IFNγ efficiently upregulated HLA-E and downregulated MICA transcript levels as we previously reported [9,35] (Figure 3A). Further, our results confirmed that both **GX** [**2**–**3** (4:6)] and **4** reduced significantly mRNA levels in IFNγ-treated ECs while **1** had no effect on MICA mRNA. **GX** [**2**–**3** (4:6)], **1** and **4** also decrease mRNA levels for STAT1 and SOCS1, two signaling proteins, which mediate IFNγ signaling pathway in the cells indicating that **GX** [**2**–**3** (4:6)], **1** and **4** are potent inhibitors of IFNγ signaling. **GX** [**2**–**3** (4:6)], **1** and **4** also provide an effective decrease in mRNA for the transcription factor CIITA pIV a master regulator of MHC molecules [4]. Since the PPAP garcinol (**4**) was previously reported as an inhibitor of histone acetyltransferase (HAT) [36] mRNA levels for HAT p300 CBP, CREB CBP and histone deacetylase HDAC3 were quantified. We found that **GX** [**2**–**3** (4:6)], **1** and **4** achieved a diminution in HATs p300 and CBP and in a minor extent in HDAC3 mRNA levels 18 h post-treatment. Interestingly, our study shows no regulatory activity, in our conditions, for simvastatin and ZA on mediators of IFNγ signaling and on CIITA expression. Consistent with these findings no regulation for HLA-E or MICA was found at the protein level (Appendix A). However, simvastatin and ZA also achieved efficient inhibition of HAT p300 and CBP.

To explore further the properties of PPAPs, the following experiments were conducted using purified xanthochymol (**2**) and guttiferone F (**3**) in comparison to garcinol (**4**) and to two synthetic inhibitors of HAT CBP/p300 (C646 and A485) and HDAC (SAHA). Firstly, we used a cell viability assay to measure the cytotoxic effect PPAPs at concentrations from 1 to 20 µM. These experiments show that PPAPs **2**–**4** and HAT or HDAC inhibitors achieved no significant cell toxicity at a concentration equal or below 10 µM (Figure 3B). Consequently, the following experiments were conducted using a concentration of 10 µM for all compounds.

The effect of purified PPAPs **2**–**4** was further investigated by qRT-PCR for the analysis of MHC, MHC-like (MICA/B), MHC-related molecules (β2-microglogulin, TAP1 and Tapasin) and proteins involved in their regulation in response to IFNγ such as STAT1, SOCS1, CIITA and IRF9 (Figure 4A). Firstly, we found that, in our experimental conditions, IFNγ increased significantly the expression of HLA class I, HLA-E, β2M, TAP1, Tapasin, STAT1, SOCS1, CIITA and IRF9. In contrast, IFNγ had no effect on MICB, GATA2 and SHP2 and decreased significantly the NKG2D ligands MICA, ULBP2 and ULBP3. In these conditions, PPAPs **2**–**4** significantly inhibited the regulatory effect of IFNγ by decreasing the transcript level for HLA class I, HLA-E, β2M and tapasin. TAP1 was selectively inhibited by **4** while **2**–**3** had no effect. PPAPs **2**–**4** further decreased MICA and MICB levels but had an opposite effect on ULBP2 and no effect on ULBP3. A significant inhibitory effect of STAT1 and SOCS1, which reflects a loss of IFNγ signal, was observed for **2** and **3** confirming the data obtained with the mixture **GX** before their purification. Next, target genes of STAT1, coding for factors involved in the transcriptional activation of MHC and HLA-E, CIITA, GATA2 and IRF9 were quantified. CIITA and GATA2 mRNA levels were dramatically reduced in the presence of PPAPs **2**–**4** consistent with an upstream dysfunction of IFNγ signaling. IRF9 was lightly increased by **2** but reduced by **4**, A485 and SAHA. No effect was found for Src homology region 2 (SH2)-containing protein tyrosine phosphatase 2 (SHP2) a protein tyrosine phosphatase involved in signal transduction by regulating several canonical pathways (MAPK and PI3K). Using Western blot experiments, we observed that PPAPs **2**–**4** reduce significantly the phosphorylation of STAT1 (Y701) and also reduce the acetylation of CBP (K1535)/p300 (K1499) suggesting hypoacetylation of CBP/p300 (Figure 4B,C). Inhibition of HAT acetylation was also achieved using A485. Together these data may suggest that, similar to garcinol (**4**), PPAPs **1**–**3** display an inhibition of HAT activities that subsequently impairs IFNγ signaling and transcriptional activity required for MHC regulation. Finally, we used flow cytometry to assess whether inhibition of HAT in EC cultures may affect MHC protein expression. To this aim, cells were treated with potent inhibitors of either HAT (C646) or HDAC (SAHA, TSA) during stimulation with IFNγ. Our findings revealed that blocking HAT using C646 dramatically impairs the expression of HLA class I and HLA-E (Figure 4D). In contrast, blocking HDACs has no effect.

### 3.4. Molecular Docking in the Binding Site of Histone Acetyltransferase p300

Biologically evaluated PPAPs **1**–**4** were also successively docked in the selective catalytic site of the histone acetyltransferase p300 previously described for A-485 [37]. Previously, the best redocked poses into the binding site (heavy atoms only) of the original structure and its energy-minimized conformation, as well as the rebuilt structure (created according to our protocol described in Materials and Methods) of the native ligand (HET code: 6TF, namely A-485) obtained with GOLD 5.6.3 deviated 0.45 Å, 0.73 Å and 1.43 Å from the true crystal pose deposited in the Protein Data Bank, respectively, denoted that the docking procedure managed to correctly pose the ligand and could be employed for further investigation (Appendix A). It was observed that the key interactions between the native ligand and several amino acid residues of the binding site were preserved in all redocked poses. Such interactions include a hydrogen bond between the A-485 ligand’s methyl-urea moiety and the backbone carbonyl of Gln1455, as well as another hydrogen bond that involves the carbonyl group substituted at the C-4′ position on the oxazolidinedione structure of A-485 and the hydroxyl group of Ser1400. Interestingly, the latter hydrogen bond was also observed in all best docked poses of PPAPs **1**–**4**. As depicted in Figure 5, for garcinol (**4**) and guttiferone F (**3**), it was the carbonyl substituent at the C-4 position of the bicyclo [3.3.1] nonane structure that participated in a hydrogen bond with Ser1400; while for guttiferone J (**1**) and xanthochymol (**2**), the carbonyl groups at C-9 (of the same cyclic scaffold) and at C-1′ were involved, respectively. As this amino acid is located at the center of the binding site, its strong interaction with the ligands could help to stabilize the protein–ligand complexes and is expected to impair the biological activity of the protein in a similar manner to that observed with the native ligand A-485.

## 4. Discussion

IFNγ is a central effector of cell-mediated immunity. Its immunomodulatory effects include enhancement of antigen processing and presentation though the regulation of MHC molecules on immune and on endothelial cells. In the present study, we demonstrated that, in the presence of IFNγ, the inhibition of MHC molecules on cell surface by the PPAPs **1**, **2**, **3** and the **GX** mixture [**2**–**3** (4:6)] was associated with reduced transcript levels for MHC molecules. This suggests that the inhibitory action of PPAPs may result from a regulation of the transcriptional activation. To decipher the transcriptional processes involved, we investigated the impact of PPAPs with a selected scaffold on the canonical IFNγ/Jak/STAT signaling pathway, which initiates the transcriptional regulation of MHC molecules.

Mechanistically, IFNγ transmits a signal through the IFNγ receptor (IFNGR), composed of IFNGR1 and IFNGR2 subunits. In canonical IFNγ-Jak-STAT1 signaling (reviewed in [38]), ligand engagement of the IFNGR leads to activation of receptor-associated kinases Jak1 and Jak2 via the phosphorylation of a receptor tyrosine residue (Y440) that serves as a docking site for STAT1, present in a latent state in the cytoplasm. STAT1 is then activated by phosphorylation of tyrosine 701 (Y701), translocates to the nucleus, binds to a regulatory DNA element termed gamma-activated sequence (GAS) and stimulates transcription of STAT1 target genes [39,40]. STAT1 binds to DNA as a dimer composed of two STAT1 subunits (α and β). Transcriptional activity of STAT1 is repressed by the negative regulators of signaling SOCS1, a key suppressor of IFNγ activities [41]. STAT1 undergoes cycles of activation–inactivation that are coupled with cytoplasmic-nuclear shuttling and regulated by post-translational modifications, including dephosphorylation of Y701 and acetylation of lysine residues K410 and K413 in the DNA binding domain (DBD) [42,43,44].

Here, we quantified the mRNA coding for transcription factors (STAT1), regulators (SOCS1) and coactivators (CIITA, GATA2 and IRF9) induced by IFNγ signaling and implicated in MHC transcription. We found that, concurrently to their effects on MHC mRNA levels, PPAPs **1**–**4** also significantly diminished mRNA for STAT1 and SOCS1, reflecting a globally reduced activity of the IFNγ signaling pathway consistent with a reduced MHC mRNA and protein expression. A decrease in the phosphorylation level of STAT1 molecules was further observed in the presence of PPAPs by Western blot, consistently with a reduced activity of the IFNγ signaling pathway. Phosphorylated STAT1 dimerized and moved rapidly into the nucleus, where it bound to the GAS element of the promoters to initiate transcription of IFNγ primary response genes. In the MHC, these included *TAP1*, Hsp70/90 (*HSPA1*), tapasin (*TAPBP*) and *CIITA* [45,46,47], which are required for subsequent activation of the HLA genes. *GATA2*, *CIITA* and tapasin were also found inhibited by PPAPs **2**–**4** while TAP1 was only inhibited by **4**. However, SHP2, which operates downstream of EGFR, dephosphorylated and inhibited *p*-STAT1 and was not inhibited by PPAPs **2**–**4** [48]. The inhibitions observed at the mRNA level for several of these intracellular factors remains to be evaluated at the protein level.

Histone hyperacetylation of MHC genes occurs rapidly after IFNγ treatment and is followed by transcription [49]. Therefore, we sought to determine whether PPAPs **2**–**4** might act initially through changes on acetylation/deacetylation shuttling as previously reported for garcinol (**4**) [50,51]. Lysine acetylation is a reversible post-translational modification that plays a crucial role in regulating protein function, chromatin structure and gene expression. Histone hyperacetylation by HATs is associated with activation of transcription, whereas HDACs is associated with transcriptional repression. Site-specific acetylation of a growing number of non-histone proteins has been shown to regulate their activity, localization, specific interactions and stability/degradation. Consequently, protein acetylation is a key target in drug design for several diseases. It was recently showed that garcinol (**4**) dose-dependently decreased the protein levels of p300/CBP HATs [52]. Thus, to characterize the biological functions of PPAPs **2** and **3**, we investigated their effect on the p300/CBP family including p300 and CREB-binding protein (CBP). CBP and p300 display dual activity as coactivators for a number of transcription factors that function as scaffolds for assembling multiprotein complexes and as enzymes that catalyze acetylation of lysine [52]. CBP and p300 HAT activity can be inhibited by NPs, such as curcumin, epigallocatechin-3-gallate and garcinol (**4**) [37]. Synthetic compounds were also identified including C646, a competitive p300 inhibitor, and A485 a CBP and p300 inhibitor with lower IC_50_ and less off-target effects [37].

Here, we indicate that xanthochymol (**2**), guttiferone F (**3**) and garcinol (**4**) decrease CBP/p300 HAT mRNA levels, suggesting that functionally both **2** and **3** display CBP/p300 HAT inhibition as **4**. In a coherent way, molecular docking experiments confirmed that PPAPs **1**-**4** might bind at the active site of p300 HAT where the native ligand A-485 is getting docked. In contrast, no significant effect was found for these compounds on the HDAC3 mRNA steady state level. Thus, since STAT1 is a non-histone target of CBP/p300 our data suggest that a reduced STAT1 mRNA level may result from the PPAPs inhibition of STAT1 acetylation/phosphorylation shuttling. Consistent with this hypothesis, our biochemical analysis showed that PPAPs efficiently decrease STAT1 phosphorylation to the same extent as A485, a specific CBP/p300 HAT inhibitor [37]. Western blots also revealed a hypoacetylation of HAT CBP/p300, a feature of HAT loss of activity [53], in the presence of PPAPs that sustains an inhibitory effect of PPAPs on HAT CBP/p300. A485 that we used as a positive control also conducted a significant hypoacetylation of CBP/p300 in our conditions. Together these findings suggest that decreased MHC expression in the presence of **2**, **3** and **4** results firstly from the post-translational inhibition of acetylation/phosphorylation of STAT1 by inhibiting HAT CBP/p300 activity that subsequently represses STAT1 transcription. Measurement of HLA-E mRNA and protein in the presence of commercial HAT inhibitors C646 and A485 further confirmed that blocking HAT CBP/p300 during IFNγ stimulation efficiently prevents the expression of HLA-E. STAT1 is required for the transcription of numerous target genes [40], several being involved in IFNγ-dependent regulation of MHC such as CIITA, IRF9, GATA2, TAP1 and Tapasin. In cultured ECs treated with IFNγ we found that **2**, **3** and **4** strongly impaired mRNA for CIITA pIV and GATA2, both being transactivators for HLA-E transcription, a key MHC molecule in ECs. A similar effect was achieved in the presence of the HAT inhibitor A485 and to a lesser extend of C646.

Our findings indicates that, by interfering with IFNγ signaling, PPAPs **1**–**4** efficiently decreased the expression of a non-classical MHC (HLA-E) and MHC-related molecules that include MICA, MICB but also proteins implicated in the peptide transport (TAP1), loading (tapasin) and MHC assembly (β2M). Concerning the decrease in HLA-E expression it can be speculated that decreased expression in the presence of PPAPs could be due directly to a specific inhibition of *HLA-E* gene transcription or could be the indirect consequence of MHC class Ia transcription inhibition leading to a lack of nonapeptides from MHC class I required for HLA-E protein expression and stability. It is also possible that both mechanisms occur. Functional differences between the two major HLA-E allelic variants (HLA-E *01:01 and HLA-E *01:03) related to HLA-E stabilization and peptide presentation have been reported [54]. Similar, MICA allelic variants also affect MICA protein function and stability [15]. Therefore, it could be of interest to further investigate the effects of PPAPs using cells homozygous for various HLA-E or MIC alleles to specifically address this point. Moreover, future experiments exploring the effect of PPAPs on cancer cells with dysregulated levels of HLA-E, MICA/B and MHC are also needed.

## 5. Conclusions

We identified guttiferone J (**1**), xanthochymol (**2**) and guttiferone F (**3**) as novel and potent inhibitors of HAT CBP/p300 activity. This is a common property shared with garcinol (**4**). Inhibition of HAT CBP/p300 activity mediated by **2** and **3** deeply impairs IFNγ signaling and ultimately MHC expression through the inhibition of STAT1 transcriptional activities probably as a result of STAT1 acetylation/phosphorylation dysregulation. Decrease in STAT1 level reduces the transcription of STAT1-dependent transactivators such as CIITA and GATA2 required for efficient transcription of MHC molecules. Overall, these findings provide new insights on the epigenetic control of MHC and propose PPAPs as useful scaffolds for improved drug design or probes targeting HAT.

## Figures and Tables

**Figure 1 biomolecules-10-01266-f001:**
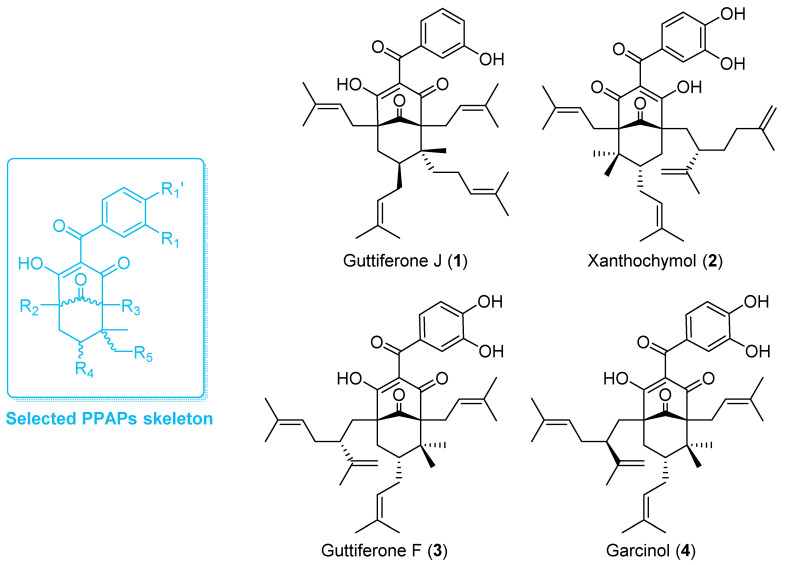
Structure of evaluated PPAPs (**1**–**4**) and their common molecular skeleton (blue).

**Figure 2 biomolecules-10-01266-f002:**
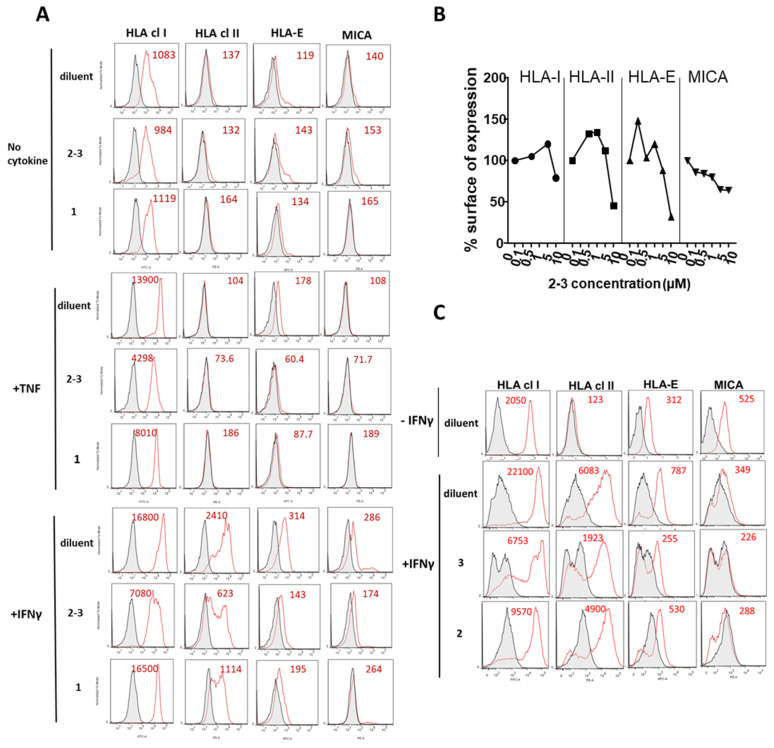
Inhibitory effect of PPAPs on major histocompatibility complex (MHC) molecules expressed on endothelial cells (ECs) during exposure to proinflammatory cytokines. (**A**) Confluent EC monolayers were incubated with either diluent (DMSO, 1/1000) only as a negative control, the mixture of PPAP **2** plus PPAP **3** [**GX** [**2**–**3** (4:6)] or with guttiferone J (**1**) at 10 µM in the presence of culture medium (no cytokine, top panel), TNF (medium panel) or IFNγ (lower panel) for 48 h. Cells were harvested, subjected to immunolabeling with specific antibodies against HLA class I, HLA class II, HLA-E and MICA and analyzed by flow cytometry. Data are depicted as histograms of fluorescence intensity (x-axis) versus cell number (y-axis) for MHC molecules (red) and for controls (irrelevant isotype control antibodies, grey). Geometric means of fluorescence are indicated in red. (**B**) Dose response of inhibition mediated by the mixture of PPAPs **2** + **3** [GX, **2**–**3** (4:6)] before purification of the two PPAPs on the expression of HLA class I, HLA class II, HLA-E and MICA and analyzed by flow cytometry. Cells were incubated with **GX** (0, 0.1, 0.5, 1.0, 5.0 and 10 µM final) in the presence of IFNγ (100 U/mL) for 48 h before immunolabeling and flow cytometry analysis. Data are expressed as percentages calculated using values obtained with cells treated with IFNγ plus diluent (DMSO 1/1000) as 100% of expression. (**C**) After purification of PPAPs, confluent EC monolayers were incubated with diluent only as a negative control, xanthochymol (**2**) or guttiferone F (**3**) at 10 µM in the absence (−IFNγ) or in the presence (+IFNγ) of IFNγ (100 U/mL) for 48 h. Cells were harvested, subjected to immunolabeling with specific antibodies against HLA class I, HLA class II, HLA-E and MICA and analyzed by flow cytometry. Data are depicted as histograms of fluorescence intensity (x-axis) versus cell number (y-axis) for MHC molecules (red) and for controls (irrelevant isotype control antibodies, grey). Geometric means of fluorescence are indicated in red. Results are representative data from 3 independent experiments.

**Figure 3 biomolecules-10-01266-f003:**
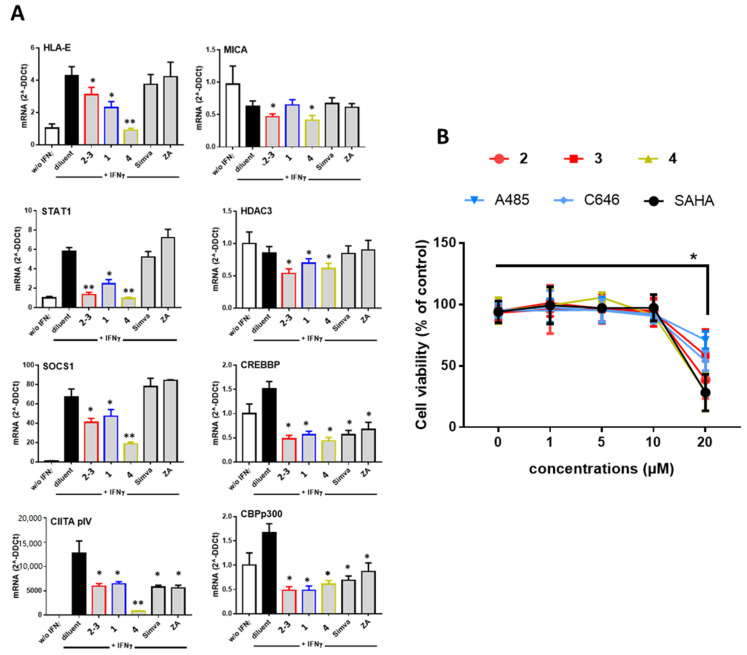
(**A**) Inhibitory effect of PPAPs on the IFNγ signaling pathway and HAT p300/HDAC3. Confluent EC monolayers were incubated with compounds **GX** [**2**–**3** (4:6)], guttiferone J (**1**) garcinol (**4**), simvastatine (Simva) or zoledronic acid (ZA) at 10 µM with IFNγ (100 U/mL) or diluent alone (DMSO, 1/1000) for 18 h. Cells were harvested for RNA isolation and qRT-PCR for the following transcripts: HLA-E, MICA, STAT1, SOCS1, CIITA pIV, CREB/CBP, p300 and HDAC3. Results shown are means of triplicate experiments, data were normalized using a housekeeping gene (RPLPO) and expressed as 2^−ΔΔCt^ using cells treated with diluent alone (DMSO, 1/1000) as reference. (**B**) Dose-dependent effect of PPAPs **2**–**4**, HAT (A485, C646) and HDAC3 (SAHA) inhibitors on cell viability. ECs were cultured for 18 h with compounds (1–20 µM) or diluent in the presence of IFNγ (100 U/mL) for 18 h. Cell viability was assessed in triplicates using MTT staining and expressed as relative percentages calculated using diluent as reference (100%), * *p* < 0.05, ** *p* < 0.01 versus diluent.

**Figure 4 biomolecules-10-01266-f004:**
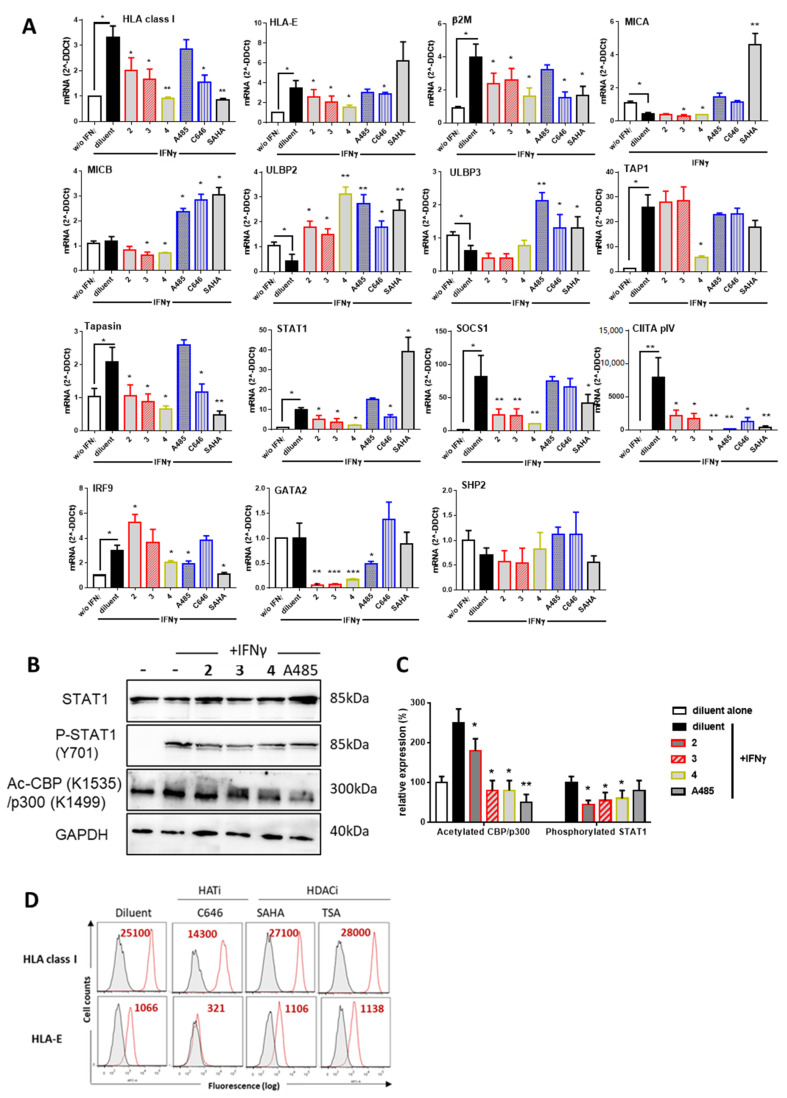
(**A**) Effect of PPAPs **2**–**4**, on MHC and MHC-related protein expression, transcriptional regulation. Confluent EC monolayers were incubated with diluent alone (DMSO, 1/1000) or with diluent, PPAPs **2**–**4**, HAT (A485, C646), HDAC3 (SAHA) inhibitors and with IFNγ (100 U/mL) for 18 h. Cells were harvested for RNA isolation and qRT-PCR for the following transcripts: HLA class I, HLA-E, MICA, MICB, ULBP2,ULBP3, β2 microglobulin (β2M), TAP1, Tapasin, STAT1, SOCS1, CIITA pIV, GATA2, IRF9 and SHP2. Results shown are means of triplicate experiments, data were normalized using a housekeeping gene (RPLPO) and expressed as 2^-ΔΔCt^ using cells treated with diluent alone (DMSO, 1/1000) as reference. (**B**) Representative Western blots for total STAT1, phosphorylated STAT1, acetylated p300/CBP and GAPDH in ECs cultured for 6 h in the presence of compounds (10 µM) and IFNγ (100 U/mL). Doublets observed for STAT1 correspond to α and β STAT1 subunits. Results are representative data from 3 independent experiments. (**C**) Quantification of immunoblots for phosphorylated STAT1, acetylated p300/CBP from 3 independent experiments after normalization to GAPDH levels. (**D**) Effect of HAT and HDAC3 inhibitors on MHC expression. Confluent ECs monolayers were cultured for 48 h with HAT (C646) or HDAC (SAHA or TSA) inhibitors at 10 µM and with IFNγ (100 U/mL) or diluent alone (DMSO, 1/1000). Cells were harvested, subjected to immunolabeling with specific antibodies against HLA class I or HLA-E and analyzed by flow cytometry. Data shown are representative histograms showing log of fluorescence intensity vs. cell number for the MHC molecules HLA class I and HLA-E (red) compared to negative controls (irrelevant isotype control antibodies, grey). Geometric means of fluorescence are indicated in red, * *p* < 0.05, ** *p* < 0.01 versus diluent plus IFNγ.

**Figure 5 biomolecules-10-01266-f005:**
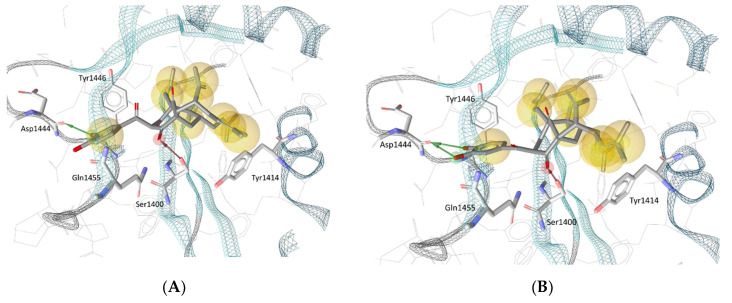
Docking poses for (**A**) garcinol (**4**), (**B**) guttiferone F (**3**), (**C**) guttiferone J (**1**) and (**D**) xanthochymol (**2**) in the binding site of histone acetyltransferase p300. Red and green arrows represent hydrogen bond acceptors and donors, respectively, and yellow spheres show the hydrophobic contacts between the ligands and the protein.

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
