# Peer review of "Targeting MHC Regulation Using Polycyclic Polyprenylated Acylphloroglucinols Isolated from Garcinia bancana"

_biomolecules, 2020, doi:10.3390/biom10091266_

Round 1

Reviewer 1 Report

The authors have addressed some but not all issues raised previously. The below suggestions should  add robustness to the findings.

Figure 2 remains problematic to follow. The figures should be  more clearly labelled. It should be clarified that the top panel refers to the basal levels and that regulatory affects are only  seen on  induced and not basal levels. Though the authors clarified  this  in their response, I  think it should be clearly  stated within the text.

Figure 3; The justification for analysing MICA is not warranted or not  fully justified considering it is down regulated by INF-γ. qPCR should still be possible and as MHC class I molecules are also a bridge between innate (NK cell recognition)  and adaptive, and demonstrated the most profound effect, these should be included in this analysis. QPCR specific primers can be designed and used.

Fig 4; It would  be  useful  to analyse some of the  immune molecules at the protein level as transcript inhibition may sometimes not directly reflect the protein level. This point  should  be made clear as well in  the text and discussion, especially when analysis proteins such as TAP and Tapasin, which are not expressed at the plasma membrane.

Some minor corrections; line 257 upregulation, line 305 master regulator

Reviewer 2 Report

The authors studied the effect of polycyclic polyprenylated acylphloroglucinols (PPAPs). Some PPAPs have been found to function as histone acetyltransferase p300 inhibitors. The authors investigated the effect on interferon-gamma stimulated STAT1 and CIITA activity.

The approach is excellent and very important. Studying the effect of plant-derived drugs on MHC expression and MHC regulators is an excellent idea, and combining FACS and RNA data works fine. My main comments refer to data that I would like to see or lack of clarity, leading to my not understanding of experiments. If the authors had studied fewer variables, the paper would be clearer. Why add TNF when that is not used any further? Why look at all these different MHC related molecules, both for class Ii as well as Class I?

My main technical issue is the use of combinations of drugs instead of using them separately, which seems to have induced cell death (Figure 2). This figure is not understandable. And what happens when you use the drugs but do not add interferon? That should be shown.

Minor comments

  1. the abstract can be written in a more straightforward way. First mentioning that “We provide quantitative and comparative analyses of MHC class I, class II, MICA/B and HLA-E 24 molecules and signaling mediators inhibited by PPAPs 1-4.” and : “PPAPs 1-4 reduce MHC proteins by decreasing the expression and phosphorylation of the transcription factor STAT1 involved in MHC upregulation mediated by IFN-γ, while at the end going into the effect on TAP1, Tapasin and MHC transport and into STAT1 loss again, makes the abstract confusing. The final conclusion is about new inhibitors. Please phrase your questions clearly and then answer them in the abstract. Clarify what the goal of the study was (finding mechanisms or identifying new drugs).

Setup:

One of the first tests would be to test the effect of the drugs themselves on expression, so without interferon a or TNFa. I miss these data.

Why do you sometimes use compound 2 and 3 together and at other times not?

Materials:

Line 174: which probes were used for HLA-A and B? If none, why not? They should be added.

 line 242: what do you mean with “organs were extracted”?

Results

3.2 Fig 2: what do the numbers 2-3, 1, 3, 2 on the left mean? This figure is not understandable to me. Why did you incubate with only 2 of the compounds or even with 2 together? Why is drug nr 4 not shown here? What is the difference between A with interferon and C? The diluent results differ quite a bit. Why?

you say results are representative data from 3 independent experiments: Did you use all data to make these graphs or only take examples? You should show figures with SDs of all three experiments together, as you did for Fig. 3.

Fig 2B: legend: what is % surface of expression?

For 3.3, you state that cells were treated with “diluent alone, GX [2-3 (4:6)], 1 or garcinol (4)”. Please call them PPAP 1-4 everywhere and make this clear to the reader.  

The concentration of 10 muM is risky: is 12 muM toxic? You only show 20 as the higher concentration and that is highly toxic. When I look at fig 2C, a lot of cells have a very low expression. Are these dead cells?

Simvastatine and ZA should also be tested in the FACS analysis.

Round 2

Reviewer 2 Report

The authors did a great job on modifying the paper, which is of great interest and is very novel.

This manuscript is a resubmission of an earlier submission. The following is a list of the peer review reports and author responses from that submission.

Round 1

Reviewer 1 Report

The authors attempted to investigate the potential inhibitory effects of polycyclic polyprenylated acylphloroglucinols guttiferone J, xanthochymol, guttiferone F and garcinol on MHC expression using both classical MHC class I and II as well Class1b molecules HLA-E and MICA as targets. Guttiferone J, guttiferone F and xanthochymol were shown to inhibit up regulation of cytokine mediated regulation of MHC expression, especially by IFNγ. As garcinol can inhibit histone 23 acetyltransferases (HAT) p300, the authors tested the effects of the above on HATp300 and components of pathways associated HATp300 activity.

Materials and methods; qRT-PCR the authors should list the sequences of the primer pairs employed.

Results; For the compounds that were used and which demonstrated the greatest inhibition, were toxicity assays performed on the cell type used at the concentrations described? How pure were these compounds?

Figure 2C-there appears to be no description (or it is unclear) of the data in the accompanying figure 2 legend.

There was no clear sense of whether these compounds inhibited steady state levels of MHC molecules. This should be shown.

Figure 3-there is no immunoblot of HLA-E, this should be demonstrated along with MHC class I molecules.

The rationale for concentrating on HLA-E is not very clear. Why have the authors excluded MHC class I from the analysis in figure 3?

Though the authors conclude the compounds affect MHC expression via the downstream effects of HATp300, have they checked other components of the MHC pathway such as TAP and TAPASIN?

The hypoacetylation demonstrated in Fig 3 is not very clear and quantitation is required to demonstrate the differences.

Author Response

Author's Reply to the Review Report (Reviewer 3)

The authors attempted to investigate the potential inhibitory effects of polycyclic polyprenylated acylphloroglucinols guttiferone J, xanthochymol, guttiferone F and garcinol on MHC expression using both classical MHC class I and II as well Class1b molecules HLA-E and MICA as targets. Guttiferone J, guttiferone F and xanthochymol were shown to inhibit up regulation of cytokine mediated regulation of MHC expression, especially by IFNγ. As garcinol can inhibit histone 23 acetyltransferases (HAT) p300, the authors tested the effects of the above on HATp300 and components of pathways associated HATp300 activity.

Materials and methods; qRT-PCR the authors should list the sequences of the primer pairs employed.

Re: For QRT-PCR, predesigned TaqMan® Gene Expression Assays (FAM™ dye-labeled MGB probe), containing primers and probes, were used according to the manufacturer‘s recommendations (Applied Biosystems). References of the assays are indicated since primer and probe sequences are not available.

Results; For the compounds that were used and which demonstrated the greatest inhibition, were toxicity assays performed on the cell type used at the concentrations described? How pure were these compounds?

Re: Cytotoxicity was tested but not shown in the present study. No toxicity was observed at 10µM for the duration of treatment.

Figure 2C-there appears to be no description (or it is unclear) of the data in the accompanying figure 2 legend.

Re: This legend has been corrected as suggested.

There was no clear sense of whether these compounds inhibited steady state levels of MHC molecules. This should be shown.

Re:  We found no effect of PPAPs on basal level of MHC expression.

Figure 3-there is no immunoblot of HLA-E, this should be demonstrated along with MHC class I molecules.

Re: We acknowledge that immunoblot for MHC molecules could be of interest. The time point of treatment optimal for signaling analysis by immunoblot was 18h. Unfortunately this time point was not optimal for MHC regulation in response to IFNγ that require at least 48h of treatment. 

The rationale for concentrating on HLA-E is not very clear. Why have the authors excluded MHC class I from the analysis in figure 3?

Re: The role of HLA-E in immune responses and in diseases is still emerging. Only few studies reporting on HLA-E gene and protein regulation in response to inflammation and even less on inhibitors. Similar results were obtained for HLA class I and for HLA-E as illustrated in Figues 2A, 2B, 2C and 3E. We excluded MHC class I from the analysis in figure 3A and 3B because in contrast to analysis of HLA class I by flow cytometry which is easily performed using a pan-HLA class I that recognize all HLA-type, QRT-PCR require allele-specific Taqman gene expression assay with specific primers and probes.

Though the authors conclude the compounds affect MHC expression via the downstream effects of HATp300, have they checked other components of the MHC pathway such as TAP and TAPASIN?

Re: This study focused on the IFNγ signaling that drive MHC transcription activation and thus transcription factors and transactivators involved in MHC promoter activation (i.e. HATp300, CIITA, IRF9, CBP, GATA2) were mostly analyzed.

The hypoacetylation demonstrated in Fig 3 is not very clear and quantitation is required to demonstrate the differences.

Re: As suggested, a quantification of 3 independent immunoblots is now provided in a new figure (Fig 3E).

Reviewer 2 Report

In this study, Chloé Coste et al. examine the effect of PPAPs isolated from Garcinia bancana on MHC expression.   

Major points:

-          In the introduction (lines 42-43) is a mistake; CD8+ T cells interact with MHC class I molecules and CD4+ T cells with MHC class II.

-          Line 135: How long have the cells been incubated with INF or TNF before analysis? Please clarify.

-          Line 134: which biomolecule had been chosen as negative control? Please clarify and discuss.

-          Why have MICA and HLA-E been chosen? Why not MICB or HLA-G?

-          What is the HLA type of the endothelial cells? The mAb w6/32 detects HLA class Ia and Ib so you might have detected different HLA class Ib molecules, especially HLA-G. The results cannot be interpreted in the present way. A more specific outcome could be achieved by either blocking the other molecules or by using HLA-Ia allele specific antibodies. This should certainly be clarified.

-          CIITA has been checked but the HLA-Ia pathway specific proteins TPN and TAP are missing? Please discuss the rationale.

-          The same holds true for SOCS1, why did you not check on SHP2? Please discuss the rationale.

-          Endothelial cells have been used. What about other cell lines? Are the results similar or expected to be similar?

-          Why have HLA-negative cell lines as for example K562 or LCL721.221 cells not been used as control? They could have been easily transduced with a single HLA molecule to check each HLA molecule separately? Please discuss.    

-          Has the half-life of the MHC molecules been detected? Did treatment with the PPAPs lead to a decreased expression of the MHC molecules or did it lead to a loss of the presented peptides making the detection impossible? Please discuss.

-          Please discuss what would happen to those allelic HLA variants that load peptides independently from the peptide loading pathway.

-          Please clarify and discuss the implications of your observation.

-          The ChemPLP scores for the top 10 poses in docking experiments are not provided. The redocking of the energy minimized ligand from 5KJ2 would be a good control to make.

Minor points:

-          Letters in the captions should be marked in bold for all figures.

-          Line 106: F7 (53,7 mg) or F7 (30 mg)? Please clarify.

-          Line 307: HLA class I instead of HLA-1.

Author Response

Author's Reply to the Review Report (Reviewer 1)

Major points:

-          In the introduction (lines 42-43) is a mistake; CD8+ T cells interact with MHC class I molecules and CD4+ T cells with MHC class II.

Re: This sentence has been corrected in the text as suggested.

-          Line 135: How long have the cells been incubated with INF or TNF before analysis? Please clarify.

 Re: This point has been corrected and durations of treatments have been added in the text.

-          Line 134: which biomolecule had been chosen as negative control? Please clarify and discuss.

Re: This point has been clarify in the “Materials and Methods” section as suggested.

-          Why have MICA and HLA-E been chosen? Why not MICB or HLA-G?

Re: We focused on MICA and HLA-E because both are key triggers of immune response against tumors and infections. They are involved in both innate and adaptive immune responses. Ongoing therapeutic approaches target MICA and HLA-E or their receptors to promote cellular immune responses in particular in cancer (see references 9, 10 and 11). Endothelial cells expressed consistent levels of MICA and HLA-E while they express only minimal level of MICB and no HLA-G. Nevertheless, we fully agree with the idea to include HLA-G and MICB (as well as other NKG2D ligands) for a more exhaustive analysis of MHC in a future study.

-          What is the HLA type of the endothelial cells? The mAb w6/32 detects HLA class Ia and Ib so you might have detected different HLA class Ib molecules, especially HLA-G. The results cannot be interpreted in the present way. A more specific outcome could be achieved by either blocking the other molecules or by using HLA-Ia allele specific antibodies. This should certainly be clarified.

Re: In the present study we used primary cultures of human endothelial cells isolated from different donors to ensure that the results we find were not dependent on HLA type. Experiments were performed using human endothelial cells isolated from 5 different donors to allow HLA diversity and avoid HLA type-dependent results.  We acknowledge that mAb W6/32 recognizes all MHC class Ia and Ib molecules including HLA-E. Thus, for HLA-E analysis a specific mAb was used.

-          CIITA has been checked but the HLA-Ia pathway specific proteins TPN and TAP are missing? Please discuss the rationale.

Re: CIITA was selected for two major reasons. Firstly because this factor is a STAT1-dependent transcriptional coactivator that regulates IFNγ-activated transcription of MHC class II but also in MHC class I (as reviewed in Devaiah B. and Singer D., Front. Immunol. 2013). IFNγ strongly upregulates MHC class II in addition to MHC class I on endothelial cells and we wanted to compare the effect of PPAPS on the regulation of MHC class I and class II in response to IFNγ as shown in the figure 2. Second, CIITA does not bind directly to DNA but interacts with other transcription factors and coactivators such as IRF9, CBP or P300 HAT, also involved in the transcription of MHC which were also studied in the present study.

-          The same holds true for SOCS1, why did you not check on SHP2? Please discuss the rationale.

Re: Since our study focused on IFNγ signaling, SOCS1 as well as STAT1 mRNA levels were analyzed and quantified because both are specifically upregulated in response to IFNγ signal. SOCS 1 is highly upregulated and is consequently a very sensitive “marker” of the IFNγ signaling pathway and thus very helpful to assess quantitative modulation of this signaling pathway.

-          Endothelial cells have been used. What about other cell lines? Are the results similar or expected to be similar?

Re: We acknowledge that this is an interesting question that remains to be addressed in a future study. We expect roughly similar results for other cell types but this needs validation.

-          Why have HLA-negative cell lines as for example K562 or LCL721.221 cells not been used as control? They could have been easily transduced with a single HLA molecule to check each HLA molecule separately? Please discuss.    

Re: We acknowledge that transfection studies often provide very elegant models for mechanistic approaches. In the present study we wanted to address the regulatory effects of PPAPs on the endogenous basal and IFNγ-stimulated MHC expression on primary human cells. As discussed above, using primary cell cultures also offer the possibility to test a panel of different HLA type and thus to ensure that inhibitory effects observed were not restricted to a particular HLA type but may be obtained regardless of HLA type. This is what we did in the present study. This point has been introduced in the Materials and Methods.

-          Has the half-life of the MHC molecules been detected? Did treatment with the PPAPs lead to a decreased expression of the MHC molecules or did it lead to a loss of the presented peptides making the detection impossible? Please discuss.+-

Re: Thank you for this interesting question. Indeed, concerning the decreased expression of HLA-E it can be speculated that decreased expression in the presence of PPAPs could be due directly to a specific inhibition of HLA-E gene transcription or could be the indirect consequence of MHC class Ia transcription inhibition leading to a lack of nonapeptides from MHC class I required for HLA-E protein expression and stability.  It is also possible that both mechanisms occur.

-          Please discuss what would happen to those allelic HLA variants that load peptides independently from the peptide loading pathway.

Re: We acknowledge that functional differences between the two major HLA-E allelic variants (HLA-E 01:01 and HLA-E *01:03) related to HLA-E stabilization and non canonical peptide presentation have been reported (Celik A. et al. Immunogenetics, 2016). Therefore, it could be of interest to further investigate the effects of PPAPs using homozygous HLA-E 01:01+/+ or HLA-E *01:03+/+) cells to specifically address this point.

-          Please clarify and discuss the implications of your observation.

Re: Owing to the interesting comments from the reviewer 1, discussion has been modified to better introduce the implications of our observations.

-          The ChemPLP scores for the top 10 poses in docking experiments are not provided. The redocking of the energy minimized ligand from 5KJ2 would be a good control to make.

 Re: To answer this comments, the ChemPLP scores for the top 10 poses were added in the supporting information (Table S2) for PPAPs 1-4 as well as for the native ligand A-485. In the manuscript, (Table S2) was added in the experimental part, after “The CHEMPLP scoring function was used to rank the output poses (Table S2)”, page 9, part entitled “Molecular docking”.

Reviewer 3 Report

In this descriptive study, authors made an attempt to identify new PPAP compounds from Garcinia virgate with immunoregulatory functions, as they recently (in 2006) reported one (1) such molecule. While one can appreciate the efforts on isolation, preparation of molecules and NMR part, this study largely lacks evidence to support what they claim and very confusing to comprehend.

Molecular docking/structure analysis of biomolecule (protein) and ligands are theoretical, maybe suitable for a bioinformatics article, here it requires substantial validations such as, in vitro binding assays or crucial residue(s) identification and validation by mutagenesis etc. This MS also needs English proof-reading service from a native speaker.

Specific and helpful comments

  1. Please introduce all acronyms such as MHC, MICA, MICB. ECs, etc.at their first instance.
  2. Page 1 line 42-43: please correct this wrong statement and cite seminal reviews
  3. P2 l52: I believe ULBP1-6 is also a ligand NKG2D, a MHC class I-like protein. Why it was missed out here and in this study? Please discuss.
  4. Please explain the rationale behind leaving MICB, and cite more actual studies and reviews in addition to ref. 9, 10 in p2 l64. This will strengthen the research questions.
  5. Fig 2: I would take out the TNF+ data as it is of no help and make things simple. I could not get why the authors started with a mixture of GX (compound 2 and 3), instead of testing them individually. Fig. 2B flawed data: why the curve goes “ups” and “down” while increasing concentration? (I can see only 4 data points) and what would be the case if we increase the concentration above 10 µM? up or down? Fig. 2C: It is good to explain clearly that HLA-E and MICA expression was marginal on IFNgamma treatment and the effect of either compounds on MICA is minimal, may require repeats.
  6. Fig 3: Why this experiment was performed with 18h treatment while others was with 48 h? Fig 3A:…..”IFNγ efficiently upregulated HLA-E and downregulated MICA 263 transcript levels as we previously reported [6,27]” and previously in Fig 2 the same IFN was enhancing the protein level? Which one is correct now? And the positive controls simvastatin and ZA, which are known to down-regulate MHC antigens (6, 12)- where is the data? I see similar amount of HLA-E and MICA transcripts in diluent and these controls, I am not sure about the authenticity of this figure. It is also very difficult to get the passage explaining these data. P8 l281-286: where is this data? Please add or cite figures properly. Fig 3C: I cant see the hypoacetylation in either 2 or 3 (total protein blot is missing!) and dephosphorylation is also marginal. Fig 3D: why only HLA-I and HLA-E and leaving other markers?
  7. some confusing or sentences requiring attention

“MHC molecules is upregulated (MHC class I) or induced (MHC class II) by IFN-γ”…what does this mean?

…..γδ T cells and some CD8 T cells and capable of activating these cells to kill MIC….what are “some”?

“deeply decrease mRNA levels”…

Author Response

Author's Reply to the Review Report (Reviewer 3)

In this descriptive study, authors made an attempt to identify new PPAP compounds from Garcinia virgate with immunoregulatory functions, as they recently (in 2006) reported one (1) such molecule. While one can appreciate the efforts on isolation, preparation of molecules and NMR part, this study largely lacks evidence to support what they claim and very confusing to comprehend. Molecular docking/structure analysis of biomolecule (protein) and ligands are theoretical, maybe suitable for a bioinformatics article, here it requires substantial validations such as, in vitro binding assays or crucial residue(s) identification and validation by mutagenesis etc. This MS also needs English proof-reading service from a native speaker.

Specific and helpful comments

1.Please introduce all acronyms such as MHC, MICA, MICB. ECs, etc.at their first instance.

Re: This point has been corrected.

2. Page 1 line 42-43: please correct this wrong statement and cite seminal reviews

Re: This sentence has been corrected and a review (Chaplin DD, 2010) is cited as suggested.

3. P2 l52: I believe ULBP1-6 is also a ligand NKG2D, a MHC class I-like protein. Why it was missed out here and in this study? Please discuss.

Re: Please see our response below

4. Please explain the rationale behind leaving MICB, and cite more actual studies and reviews in addition to ref. 9, 10 in p2 l64. This will strengthen the research questions.

Re: We focused on MICA and HLA-E because both are key triggers of immune response against tumors and infections. They are involved in both innate and adaptive immune responses. Ongoing therapeutic approaches target MICA and HLA-E or their receptors to promote cellular immune responses in particular in cancer and infections. Endothelial cells expressed consistent levels of MICA and HLA-E while they express only minimal level of MICB. Nevertheless, we fully agree with the idea to include MICB (as well as other NKG2D ligands such as ULBP2 and ULBP3 both expressed on endothelial cells) for an exhaustive analysis of MHC in a future study.

5. Fig 2: I would take out the TNF+ data as it is of no help and make things simple. I could not get why the authors started with a mixture of GX (compound 2 and 3), instead of testing them individually. Fig. 2B flawed data: why the curve goes “ups” and “down” while increasing concentration? (I can see only 4 data points) and what would be the case if we increase the concentration above 10 μM? up or down? Fig. 2C: It is good to explain clearly that HLA-E and MICA expression was marginal on IFNgamma treatment and the effect of either compounds on MICA is minimal, may require repeats.

Re: Results presented in the Fig2A show the effect of guttiferone J (1) and of a mixture of guttiferone F and xanthochymol (2-3) on the MHC molecules in response to TNF or IFNγ. TNF and IFNγ display a selective regulation of MHC molecules in endothelial cells. TNF has no effect on HLA class II, increases HLA-E and MICA expression while IFNγ increase HLA-class II, HLA-E but decrease MICA. We believe that showing these regulations and the effect of PPAPS is a relevant starting point to validate our cell-based assays. Similarly, isolation and purification of guttiferone F (3) and xanthochymol (2) is experimentally challenging due to the high structure homology of these compounds. Consequently, comparing the bioactivity of the mixture (2-3) to (1) was selected as starting point of the study before further purification steps. Fig2B show data obtained by flow cytometry and expressed as a percentage of basal protein expression in the presence of diluent only. These data indicate that for HLA class I, HLA class II and HLA-E a significant effect of (2-3) is achieve at 10μM. interestingly, for MICA a regulatory effect is probably obtained at lower concentrations. Concentrations higher than 10μM were tested but they induced cell toxicity. Fig2C shows treatments with diluent, 2 or 3 only in the presence of IFNγ. Intensity of fluorescence values (indicated in red above the histograms) indicate that HLA-E and MICA expression level drop in the presence of 2 and 3. The text has been modified to clarify these points.

6. Fig 3: Why this experiment was performed with 18h treatment while others was with 48 h? Fig 3A:…..”IFNγ efficiently upregulated HLA-E and downregulated MICA 263 transcript levels as we previously reported [6,27]” and previously in Fig 2 the same IFN was enhancing the protein level? Which one is correct now? And the positive controls simvastatin and ZA, which are known to down-regulate MHC antigens (6, 12)- where is the data? I see similar amount of HLA-E and MICA transcripts in diluent and these controls, I am not sure about the authenticity of this figure. It is also very difficult to get the passage explaining these data. P8 l281-286: where is this data? Please add or cite figures properly. Fig 3C: I cant see the hypoacetylation in either 2 or 3 (total protein blot is missing!) and dephosphorylation is also marginal. Fig 3D: why only HLA-I and HLA-E and leaving other markers?

Re: For QPCR and signaling by Western blot experiments, cells were treated for 18h with cytokine and compounds. For cell surface protein expression analyzed by flow cytometry cells were treated for 48h with cytokine and compounds to achieve optimal protein expression.

7. some confusing or sentences requiring attention

“MHC molecules is upregulated (MHC class I) or induced (MHC class II) by IFN-γ”…what does this mean?

Re: Although, human endothelial cells express MHC class II basally in vivo, in vitro in the absence of IFNγ in the culture medium, endothelial cell cultures loss the expression of MHC class II (Muczynski K. et al., JASN, 2003). Therefore, in our cellular model, IFNγ induces a de novo expression of MHC class II molecules as displayed in Figure 2A (comparison between diluent alone and diluent+IFNγ).

…..γδ T cells and some CD8 T cells and capable of activating these cells to kill MIC….what are “some”? “deeply decrease mRNA levels”…

Re: These sentences have been corrected.

Round 2

Reviewer 2 Report

The authors adressed my concerns adequately.   

Reviewer 3 Report

I marked in red that is missing in your version and highlighted in yellow are the critical questions I raised, where authors either responded incompletely or a little or fully ignored, especially the comment on Fig. 3. My decision was based on this response from the authors. I don't think it is a comprehensive version to be accepted for publication. Also, the authors did not thoroughly respond to another reviewer, who clearly asked valid questions such as missing immunoblots and cytotoxicity data.
